## [Decision Letter · Decision Letter 0]

4 Aug 2025

Is working from home changing the meaning of work?

PLOS ONE

Dear Dr. Collischon,

Thank you for submitting your manuscript to PLOS ONE. After careful consideration, we feel that it has merit but does not fully meet PLOS ONE’s publication criteria as it currently stands. Therefore, we invite you to submit a revised version of the manuscript that addresses the points raised during the review process.

We look forward to receiving your revised manuscript.

Kind regards,

Prof. Katarzyna Piwowar-Sulej

Academic Editor

PLOS ONE

Journal Requirements:

Reviewers' comments:

Reviewer's Responses to Questions

**Comments to the Author**

1. Is the manuscript technically sound, and do the data support the conclusions?

Reviewer #1: Yes

Reviewer #2: Partly

2. Has the statistical analysis been performed appropriately and rigorously?

Reviewer #1: No

Reviewer #2: Yes

3. Have the authors made all data underlying the findings in their manuscript fully available?

Reviewer #1: Yes

Reviewer #2: No

4. Is the manuscript presented in an intelligible fashion and written in standard English?

Reviewer #1: No

Reviewer #2: Yes

Reviewer #1: manuscript introduces Jahoda’s latent deprivation model but does not delve deeply into the psychological theories underlying it. Consider extending the theoretical framework and connecting it with contemporary perspectives in occupational or organizational psychology.

2. The binary classification of WFH (always, never) is too simplistic. It misses the nuance in frequency (1 day/week vs. 5 days/week), voluntariness, and hybrid models. Collecting or leveraging more detailed data on WFH intensity would enhance explanatory power.

3. Most results are null, but there’s insufficient discussion about statistical power. Are the sample sizes in subgroups (parents, managers) large enough to detect small-to-moderate effects? Include formal power calculations or at least a sensitivity analysis.

4. Although the paper acknowledges potential self-selection into WFH, it does not explicitly model it. Consider using techniques such as instrumental variables, matching, or Heckman correction if plausible instruments or selection models are available.

5. The annual survey waves don’t capture short-term fluctuations in WFH experiences or outcomes. This temporal mismatch may attenuate observed effects. The authors should discuss this limitation more rigorously and suggest finer-grained data collection for future studies.

6. The significant finding for higher social integration among those who always WFH is speculated to be due to unobserved heterogeneity. Instead of speculation, test for such heterogeneity more systematically (by using interaction terms or stratified models).

7. The effects of WFH likely vary by sector (e.g., tech vs. healthcare) or job function (creative vs. administrative). The study could benefit from stratified analysis or examination of interaction effects by occupation or industry.

8. While job satisfaction is included, other aspects of job quality (autonomy, perceived fairness, career development) are not directly addressed. Consider adding these variables or discussing why they were excluded.

9. The manuscript claims results are likely generalizable, but cultural, regulatory, and economic differences may moderate WFH’s effects. Include more comparative references or theoretical discussion on cross-national variability.

10. Many subgroup analyses are relegated to the appendix without interpretation in the main body. Consider integrating key findings more directly into the results and discussion sections to enhance reader engagement.

11. The paper does not explore mediators (e.g., perceived autonomy, work-life balance) or moderators ( gender, personality traits). Including these could shed light on the mechanisms or boundary conditions of WFH effects.

12. Some sections (the introduction) jump between themes, such as environmental impacts and employee well-being, without clear transitions. Streamlining the narrative around core research questions would help focus the reader.

Reviewer #2: The paper is a well-structured and clearly written paper on a subject of broad interest. The null results are potentially valuable, especially given the shift toward hybrid and remote work. However, the study could be significantly improved by addressing the self-selection issue and moving closer to a causal framework. With these revisions, the paper would offer a more robust contribution to the literature on remote work and worker well-being. I suggest the authors to revise and resubmit the manuscript.

**Do you want your identity to be public for this peer review?** For information about this choice, including consent withdrawal, please see our Privacy Policy

Reviewer #1: No

Reviewer #2: **Yes:** Michela Bia

---

## [Author Response · Author response to Decision Letter 1]

29 Sep 2025

See the attached file for the responses to the referees' comments.

---

## [Decision Letter · Decision Letter 1]

26 Nov 2025

PONE-D-25-30593R1Is working from home changing the meaning of work?PLOS ONE?

Dear Dr. Collischon,

Thank you for submitting your manuscript to PLOS ONE. After careful consideration, we feel that it has merit but does not fully meet PLOS ONE’s publication criteria as it currently stands. Therefore, we invite you to submit a revised version of the manuscript that addresses the points raised during the review process.

After a careful evaluation of the submitted manuscript, I must note that the document in its current form does not meet the standards of a scientific article. Instead, it resembles a research report. Substantial revisions are necessary before the manuscript can be considered for publication. Below I outline the key issues.

**Lack of academic article structure**he *Introduction* fails to identify a clear **research gap** . It remains unclear what has already been studied in the field and what specific gap the present study aims to address. In journals such as *PLOS ONE* , authors are required to define the context, the problem, and the research gap clearly so that the contribution of the article is explicit.**Missing theoretical framework**The manuscript does not follow the conventional structure (Introduction,**Literature review** , Methods, Results, Discussion). Most importantly, tthe manuscript does not contain a theoretical section that would allow the development of hypotheses or research questions. **As a result, the empirical part appears disconnected from any conceptual background.**A valid empirical study must be grounded in theory; otherwise, the rationale behind the research design and hypotheses remains unclear.**Insufficient discussion of findings**The **Discussion section should compare the findings with previous research** . In the current version, the authors do not relate their results to similar studies conducted by other scholars.Moreover, the manuscript fails to articulate its **contribution to theory and practice** in the field of management. This must be presented explicitly and supported with argumentation.

As a reference, I encourage the authors to review recently published articles in *PLOS ONE* , such as:

**
https://doi.org/10.1371/journal.pone.0335751
**

This will help illustrate the expected structure, depth of theoretical grounding, and clarity of contribution.

We look forward to receiving your revised manuscript.

Kind regards,

Katarzyna Piwowar-Sulej

Academic Editor

PLOS ONE

---

## [Author Response · Author response to Decision Letter 2]

19 Dec 2025

The response to the editor's comments on the previous submission are explained in detail in the corresponding attachment to this submission.

---

## [Editor Report · Decision Letter 2]

22 Dec 2025

Is working from home changing the meaning of work?

PONE-D-25-30593R2

Dear Dr. Collischon,

We’re pleased to inform you that your manuscript has been judged scientifically suitable for publication and will be formally accepted for publication once it meets all outstanding technical requirements.

Kind regards,

Katarzyna Piwowar-Sulej

Academic Editor

PLOS One

---

## [Editor Report · Acceptance letter]

PONE-D-25-30593R2

PLOS One

Dear Dr. Collischon,

I'm pleased to inform you that your manuscript has been deemed suitable for publication in PLOS One. Congratulations! Your manuscript is now being handed over to our production team.

Kind regards,

on behalf of

Professor Katarzyna Piwowar-Sulej

Academic Editor

PLOS One